# Aquaporins in Biliary Function: Pathophysiological Implications and Therapeutic Targeting

**DOI:** 10.3390/ijms252212133

**Published:** 2024-11-12

**Authors:** Mohamad Khalil, Patrizia Gena, Agostino Di Ciaula, Piero Portincasa, Giuseppe Calamita

**Affiliations:** 1Clinica Medica “A. Murri”, Department of Precision and Regenerative Medicine and Ionian Area (DiMePre-J), University of Bari “Aldo Moro”, 70121 Bari, Italy; mohamad.khalil@uniba.it (M.K.); agodiciaula@gmail.com (A.D.C.); 2Department of Biosciences, Biotechnologies and Environment, University of Bari “Aldo Moro”, 70125 Bari, Italy; annapatrizia.gena@uniba.it

**Keywords:** bile, gallbladder, gallstone disease, liver, transmembrane flux of water

## Abstract

Aquaporins (AQPs) are transmembrane proteins permeable to water and a series of small solutes. AQPs play a key role in pathways of hepatobiliary secretion at the level of the liver, bile ducts, and gallbladder. AQP8 and -9 are pivotal in facilitating the osmotic water movement of hepatic bile, which is composed of 95% water. In the biliary tract, AQP1 and -4 are involved in the rearrangement of bile composition by mechanisms of reabsorption/secretion of water. In the gallbladder, AQP1 and -8 are also involved in trans-epithelial bidirectional water flow with the ultimate goal of bile concentration. Pathophysiologically, AQPs have been indicated as players in several hepatobiliary disorders, including cholestatic diseases and cholesterol cholelithiasis. Research on AQP function and the modulation of AQP expression is in progress, with the identification of potent and homolog-specific compounds modulating the expression or inhibiting these membrane channels with promising pharmacological developments. This review summarizes the contribution of AQPs in physiological and pathophysiological stages related to hepatobiliary function.

## 1. Introduction

Bile is a dark green to yellowish-brown liquid found in the liver, gallbladder, and intestine and made mainly of water and solutes. Among the solutes, there are three main lipid species in bile which are cholesterol, phospholipids, and bile acids (BAs) [1]. Additional solutes are bilirubinate pigments, small amounts of proteins, and inorganic salts [2]. Proteins and metabolites of a series of endogenous compounds such as hormones are present in minor concentrations [3,4]. Major biliary inorganic ions are Na^+^, K^+^, Ca^++^, Mg^++^, Cl^−^, and HCO_3_^−^, whose concentrations in the common duct bile are quite close to the plasmatic ones. After liver production and secretion, diluted bile enters the gallbladder where it is concentrated during fasting. Following a fat-enriched meal and neurohormonal stimulus leading to gallbladder contraction, excreted bile enters the duodenum and flows across the intestine. Here, bile plays a key role in digestion with increasing emulsification and absorption of dietary lipids (e.g., triglycerides, cholesterol) and fat-soluble vitamins, e.g., A, D, E, and K. In addition, bile is the main route for the excretion of cholesterol from the body, with BAs and phospholipids composed of >95% lecithins acting as cholesterol carriers. This process of aggregation starts at the canaliculus side of hepatocytes by forming simple/mixed micelles and unilamellar/multilamellar vesicles according to their luminal concentrations [5].

Aquaporins (AQPs) are membrane proteins that facilitate the movement of water and some small neutral solutes across cell membranes [6] (Figure 1). Mammalian AQPs are composed of 13 members (AQP0 to AQP12), of which some are mostly permeable to water (orthodox aquaporins: AQP0, AQP1, AQP2, AQP4, AQP5, AQP6, AQP8), while the others are also permeable to glycerol (aquaglyceroporins: AQP3, AQP7, AQP9, AQP10) and/or hydrogen peroxide (peroxiporins: AQP1, AQP3, AQP5, AQP8, AQP9 and AQP10, AQP11). A fourth group is composed of two members, AQP11 and AQP12, which are often indicated as superaquaporins or unorthodox aquaporins due to their divergent evolutionary pathway, lower primary sequence homology, transport properties, and subcellular localization [7,8]. This classification is not exhaustive since AQPs also allow the transmembrane movement of urea (AQP3, AQP6, AQP7, AQP9, AQP10), nitric oxide (AQP1, AQP4), ammonia (AQP1, AQP3, AQP6, AQP7, AQP8, AQP9), and gases such as oxygen (AQP1) and carbon dioxide (AQP0, AQP1, AQP4, AQP5, AQP6, AQP9) [9]. Moreover, some AQPs can mediate the diffusion of some inorganic ions (AQP0, AQP1, AQP6), silicon (AQP3, AQP7 AQP9, AQP10), and antimonite and arsenite (AQP7, AQP9) [10].

As shown in Figure 1, AQPs play a crucial role in maintaining metabolic homeostasis, with their distribution, expression, and function being regulated under both normal and pathological conditions. AQPs are involved in various physiological processes across multiple organ systems, and their dysregulation has been implicated in a range of diseases [11]. These include metabolic syndrome [12], cardiovascular diseases [13], renal concentration disorders [14], obesity [15], diabetes [16], liver steatosis [17], and gallstones [18]. Furthermore, AQPs are linked to cancer, inflammation, and a wide range of diseases [19,20,21,22,23].

In the hepatobiliary system, AQPs are expressed in various cell types. Hepatocytes, the main cells of the liver, express AQP8, AQP9, and AQP11. AQP8 features multiple subcellular localization and plays roles in bile water secretion, ammonia detoxification, redox balance, and cholesterol biosynthesis. AQP9 is at the basolateral plasma membrane and is involved in the uptake of glycerol, urea, and other solutes [24,25]. Cholangiocytes, the epithelial cells lining the bile ducts, are responsible for ductal bile secretion. AQP1 is expressed in cholangiocytes and is involved in water flow during bile formation. Secretin, an intestinal hormone, regulates AQP1-facilitated water secretion under cAMP signaling [26]. The gallbladder stores and concentrates bile. The gallbladder epithelial cells express AQP1 and AQP8, which contribute to the water movement into and out of the gallbladder [27].

Dysregulated AQPs have been observed in various liver and biliary diseases, and this review discusses the physiological implication of various AQPs during bile secretion and reabsorption. Both natural and synthetic compounds are objects of study for their effects on increased or decreased expression and channel function of AQPs. This approach contributes to unraveling the pathophysiological regulation of biliary water transport. Research in this area is ongoing to target AQPs in biliary diseases with the help of agonists, antagonists, gene therapies, and more. Although preclinical studies have already shown promising results, the research must take into account factors such as disease mechanisms, desired therapeutic effect, safety, and efficacy.

Based on such assumptions, here we provide an updated and comprehensive review of key pathophysiological aspects of AQPs’ function in health and disease and provide insights on potential and novel strategies to modulate AQPs as additional therapeutic targets.

## 2. The Role of AQPs in Bile Flow

Bile is made of ~95% water [5,28], and in healthy conditions, adult humans secrete about 0.8–1.0 L of hepatic bile daily at a rate of 30–40 mL per hour. Bile production is about six times higher in rats, a species lacking the gallbladder [1]. Bile fluid formation begins at the canalicular (apical) membrane of hepatocytes as an osmotic process involving solutes and water. BAs and other biliary constituents are actively secreted into bile canaliculi, creating the osmotic force that drives the parallel water secretion [1]. BA-independent bile flow also exists, since canalicular bile flow is also found in the absence of BAs or at low bile acid outputs. This additional bile flow component is attributed to the active secretion of osmotically active inorganic electrolytes and organic anions. Total bile formation results from constant ductal secretion and total canalicular bile flow, consistent with the linear relation existing in both total bile flow and total canalicular bile flow [1].

The process of the enterohepatic circulation of bile with its solutes is a complex one and is mainly driven by the BA pool. The “primary” BAs synthesized in the liver from cholesterol are trihydroxy cholic acid (CA) and dihydroxy chenodeoxycholic acid (CDCA). About 5% of primary BAs escape ileal absorption and enter the colon, where the resident microbiota initiates BA deconjugation from taurine and glycine, dehydrogenation, dehydroxylation, and epimerization to produce “secondary” BAs: dihydroxy deoxycholic acid (DCA) and monohydroxy lithocholic acid (LCA). The 7α-dehydrogenation of CDCA forms dihydroxy 7α-oxo (keto)-LCA, which is metabolized to the “tertiary” 7β-epimer, dihydroxy ursodeoxycholic acid (UDCA), in the colon and to CDCA again in the liver. LCA, 7-oxo (keto)-LCA, and UDCA are mainly excreted in feces, while about 50% of DCA is passively reabsorbed from the colon into the portal tract [29] by ionic more than nonionic diffusion (the remaining part being excreted with feces) [30]. Altogether, the BA pool in every cycle undergoes reconjugation with taurine and glycine and new secretion in bile. Fecal loss is minimal (<5% in every cycle). As an example, when a CA or CDCA pool of 1 g cycles six times a day, the daily loss is 5% × 6 cycles = 30%, and 300 mg must be resynthesized in the liver [31].

The epithelial cells lining the mammalian hepatobiliary tree express several AQPs with distinct subcellular localizations and roles (Table 1). As in the blood vessels of other body districts, the endothelial cells of the hepatobiliary system express AQP1 [32].

### 2.1. Role of AQPs in the Hepatobiliary Tract

#### 2.1.1. Liver

Human and rodent hepatocytes express high levels of AQP8 and AQP9. Mouse hepatocytes also express AQP11 [33], which was also found in Huh-7 cells, an immortalized human hepatocyte cell line [34]. Human hepatocytes also express AQP3 and AQP7. The redundancy of AQPs in human hepatocytes is still not clear, although it is reasonable to think that it is justified by the different molecular selectivity and subcellular localization that distinguishes them [24]. However, while the roles attributed to AQP8, AQP9, and AQP11 are important [35], the relevance of AQP3 and AQP7 in hepatocytes, if they have physiological meaning at all, is unclear.

In hepatocytes, AQP8 features multiple subcellular localizations that range from the canalicular membrane and subapical vesicles to organelles such as the mitochondria and the smooth endoplasmic reticulum [36,37]. Hepatocyte AQP8 is reported to have multiple functions, including mediating the secretion of canalicular bile water [18,38,39], preserving the cytoplasm osmolarity during glycogen synthesis or degradation [37], facilitating the ammonia movement in mitochondrial ammonium detoxification and ureagenesis [40,41,42,43], and mediating the efflux of hydrogen peroxide out of mitochondria during oxidative stress [44,45]. Studies with human Huh-7 cells and primary rat hepatocytes suggested a role for mitochondrial AQP8 in hepatocyte cholesterol biosynthesis modulated through the sterol regulatory element-binding protein (SREBP) [46,47,48]. A recent work with obese mice showed hepatic AQP8 overexpression leading to activation of the farnesoid X receptor (FXR) with the inhibition of genes related to lipogenesis and further diminution of intrahepatic triacylglycerol (TAG) overaccumulation [49]. Based on its peroxiporin property, AQP8 has been suggested to be involved in the differential regulation of metabolic signaling by α1- and β-adrenoceptors (ARs) with consequent induction of Ca^2+^ ion mobilization. Hydrogen peroxide inhibited the β-AR-mediated activation of the glycogenolytic, gluconeogenic, and ureagenic responses induced by α_1_-AR, the NOX2-H_2_O_2_-AQP8-Ca^2+^ signaling cascade, an observation that led to the hypothesis that the signaling related to the H_2_O_2_ moving through AQP8 is an additional pathway acting downstream of α_1_-AR in hepatocytes. The inhibitory effect of hydrogen peroxide on β-AR signaling may also explain the negative crosstalk existing between the two signaling pathways [50]. Extensive work has been carried out in deciphering the role of AQP8 in canalicular bile secretion [24,38]. Choleretic agonists such as dibutyryl cyclic adenosine monophosphate (cAMP) and glucagon were shown to trigger the translocation of subapical vesicles incorporating AQP8 to the canalicular plasma membrane through a phosphatidylinositol-3-kinase-dependent microtubule-associated pathway [51]. Insertion of AQP8 in the hepatocyte apical membrane was accompanied by an increase in the water permeability of the canalicular plasma membrane and osmotic movement of water into the bile canaliculus [18,39,52,53] (Figure 2).

A similar observation was made in a study using rat primary hepatocytes where glucagon increased the AQP8 protein level by diminishing its degradation through a process involving the cAMP-PKA and PI3K signal pathways [54]. Cyclic AMP-induced redistribution from the cytoplasmic compartment to the apical membrane of hepatocytes also occurs for molecular carriers participating in canalicular bile secretion, such as isoform 2 of the Cl^−^/HCO_3_^−^ exchanger (AE2) and multidrug resistance-associated protein 2 (MRP2). The implication of AQP8 in hepatic bile formation was argued in a study with hepatocytes from *Aqp8*^−/−^ mice where the canalicular osmotic water permeability was reported to be of a similar extent to that of *Aqp8*^+/+^ wild-type mice [55]. This apparent discrepancy could be explained by the redundancy of AQPs in hepatocytes and by the fact that the osmotic water permeability of the hepatocyte canalicular membrane is not completely linked to AQP8. Regarding this last aspect, a study with rat primary hepatocytes showed that the water permeability of the canalicular membrane does not entirely depend on AQP8 since a 60% decrease in the AQP8 level in the apical membrane led to a 15% reduction in the overall canalicular osmotic permeability [56]. Furthermore, the paradigm that the water flow into the canaliculus depends only on the rate of carrier-mediated transport was recently challenged by the changes in hepatic bile composition seen in the Claudin-2 knockout mouse and after examining the cholestatic effect of estradiol 17*β*-d-glucuronide [57]. A respective decrease in paracellular or transcellular canalicular water flow, likely through AQP8, did not show any significant effect on bile acid excretion. Useful information to unravel this aspect is provided by a recent work with *Aqp8*^−/−^ knockout mice showing reduced canalicular bile formation, characterized by the secretion of concentrated bile with a lower flow rate and higher levels of bile lipids than that of *Aqp8*^+/+^ littermates [18].

AQP9 is an aquaglyceroporin of broad selectivity that, besides water, allows movement of a wide variety of neutral solutes including glycerol and other polyols, H_2_O_2_, urea, carbamides, nucleosides, monocarboxylates, purines, pyrimidines, and metalloid arsenic [58,59]. Multiple functions have been ascribed to this aquaglyceroporin [25]. Both in rodent and human hepatocytes, AQP9 is localized at the sinusoidal domain of the basolateral plasma membrane [60]. In rodents, AQP9 is the principal pathway through which glycerol is taken up by hepatocytes from the portal blood during fasting [61,62]. Imported glycerol is promptly converted into glycerol-3-phosphate (G3P), a major substrate for gluconeogenesis in early starvation. AQP9 also has relevance in lipid homeostasis since G3P is needed for triacylglycerol synthesis [63]. Hepatocyte AQP9 is also suggested to be involved in rodent bile formation and in extruding catabolic urea by facilitating the entry of water and the exit of urea from and to portal blood, respectively [64,65]. In rodents, the expression of AQP9 is negatively regulated by insulin at the transcriptional level [66], explaining why hepatic AQP9 is enhanced in situations of insulin resistance [67,68]. The functional relevance of AQP9 in metabolic homeostasis and energy balance was also indicated by studies with AQP9-depleted knockout mice, where the lack of AQP9 was associated with diminished liver glycerol permeability and enhanced levels of plasma glycerol and TAGs [65,69]. Reduced levels of AQP9 occurred in hepatocytes of murine models of obesity and subjects with obesity with type 2 diabetes where a significant reduction in liver glycerol permeability was also found [70,71]. The hepatic expression of AQP9 is also influenced by leptin [63,72], a peptide hormone that is produced in fat cells, the placenta, and, to a lesser degree, the gut. Leptin reflects both energy stores, e.g., predominantly fat, and energy balance, e.g., weight loss or maintenance, and binds to leptin receptors on the surface of neurons in the hypothalamus to modulate food intake and energy balance [73]. However, the modulation exerted by both insulin and leptin on AQP9 appears to be different between humans and rodents [35], an aspect that deserves further investigation. Sex-related dimorphism of liver AQP9 expression is reported in both rodents and humans, in line with the known distinctions between the two genders in handling glycerol for metabolic purposes [72,74]. Sex-specific differences are also found for AQP3 and AQP7, two other AQPs of metabolic significance in adipose tissue [72,75]. Using a rat hepatoma cell line, AQP9 showed a role in the lipid-lowering activity of silybin, a nutraceutical phytocompound, through modulation of the autophagic process and lipid droplet composition [76]. Liver AQP9 is also reported to be of strong immune relevance. TLR4 ligands such as LPS have been reported to involve AQP9 in the mechanism leading to the production of inflammatory NO and O_2_^−^ through the involvement of the NF-kB pathway [77] and NLRP3 inflammasome [78]. A subsequent study showed that the inhibition of AQP9 with the specific and potent blocker RG100204 abolishes the LPS-induced increase in NO and O_2_^−^ in FaO cells, a rat hepatoma cell line [79]. However, further work is needed to better understand the role of hepatic AQP9 in the immune system since a recent study reported that mice with gene knockout of *Aqp9* recruit B and CD4^+^ T cells through the upregulation of cathepsin S and macrophage activation with the induction of the immune and inflammatory responses [80].

Mouse liver AQP11 has been suggested to contribute to rough endoplasmic reticulum homeostasis and liver regeneration, although the exact mechanism with which this superaquaporin intervenes is unclear [34,81]. The recent functional identification of AQP11 as a peroxiporin opens new horizons about its possible involvement in intracellular H_2_O_2_ homeostasis and ER stress prevention [8]. AQP11 has been suggested to mediate the transfer of H_2_O_2_ from mitochondria to the endoplasmic reticulum as an interorganellar redox response activated upon the downregulation of endoplasmic reticulum flavoenzyme oxidoreductin-1 alpha (ERO1α) [82]. Further studies are therefore expected to assess the role of AQP11 in the liver.

#### 2.1.2. Bile Ducts

Cholangiocytes, the epithelial cells lining the lumen of the biliary tree, are responsible for secretin-induced ductal bile secretion through a cAMP-dependent pathway [83] and activation of Cl^−^ efflux via cystic fibrosis transmembrane conductance regulator (CFTR) that drives the extrusion of HCO_3_^−^ into the lumen via apical AE2 (chloride/bicarbonate exchanger). Both HCO_3_^−^ and Cl^−^ represent the main driving force for the osmotic movement of water through apical AQP1 into the biliary lumen [83,84]. Human and rodent cholangiocyte AQP1 mediates the apical secretion of water during both basal- and hormone-regulated ductal bile formation [32,85,86]. AQP1 is also present in subapical membrane vesicles [87] in co-expression with AE2 and CFTR [88], and secretin was reported to regulate the exocytic insertion of these vesicles into the apical membrane of cholangiocytes [35,87]. This brought forth a novel paradigm of the functional bile secretory unit. Cholangiocytes express AQP4 and AQP1 at the basolateral plasma membrane domain [87,89]. AQP-mediated water movement is believed to permit the relative isosmolar status maintained during ductal bile formation. This is consistent with the physical association existing between the basolateral membrane of cholangiocytes and the peribiliary vascular plexus surrounding the bile ducts and from which bile water originates [24,90] (Figure 3).

However, the water permeability of cholangiocytes isolated from AQP1-depleted knockout mice did not decrease [91]. Compensatory upregulation of other mouse cholangiocytes AQPs (e.g., AQP8) was hypothesized to explain this unexpected observation [92,93]. Intrahepatic bile ducts not only secrete but also absorb water, as proved in isolated rodent intrahepatic bile duct units [94]. Likely, the osmotic absorption of water is triggered by the active absorption of sodium-coupled glucose and BAs through the SGLT1 and ASBT cotransporters, respectively [83]. Somatostatin, gastrin, and insulin, hormones that decrease the intracellular levels of cholangiocyte cAMP, may act by inhibiting the secretin-induced vesicular transport of AQP1, CFTR, and AE2 to the apical membrane of cholangiocytes by decreasing the ductal bile secretion [95]. This mechanism may also explain why somatostatin reduces ductal secretion while stimulating the net absorption of ductal water. After all, the functional association between CFTR and AQPs has been described in murine Sertoli cells [96,97].

#### 2.1.3. Gallbladder

The mammalian gallbladder represents the dynamic reservoir of diluted hepatic bile flowing bidirectionally across the cystic duct. The periodic daily fluctuations made of gallbladder emptying and refilling episodes are governed by neurohormonal stimuli and play a key role in lipid digestion and metabolic homeostasis, along with the enterohepatic circulation of BAs [5,30,98]. The concentration of stored bile occurs during fasting and depends on the movement of water across the gallbladder epithelium. This step is driven by osmotic gradients generated from active salt absorption and secretion [24,99]. Both human and murine gallbladder epithelial cells express AQP1 and AQP8. AQP1 is found both at the apical and at the basolateral plasma membrane of the epithelial cells lining the neck of the organ [100]. AQP1 is also found at the corpus portion of the gallbladder where immunoreactivity has been observed at the plasma membrane and over subapical vesicles that can be incorporated into the apical membrane by means of a microtubule-dependent, cAMP-stimulated mechanism whose stimulation is not yet known [101]. In murine gallbladder, AQP1 was reported to be slightly upregulated by leptin [102]. AQP8 has been described at the plasma membrane and, to a lesser extent, at intracellular vesicles of the gallbladder epithelium of different species [32,36]. A recent work reported the upregulation of gallbladder cholangiocytes AQP1 and AQP8 and of CFTR by the liver X receptor β (LXRβ), an oxysterol-activated transcription factor strongly expressed in the gallbladder epithelium [27]. A molecular partnership between CFTR and AQPs has also been found in mouse Sertoli cells [97]. Very high water permeability (*P*_f_ of 0.2 cm/s) in mouse gallbladder epithelium involving transcellular water transport through AQP1 was found in a study using wild-type mice [103]. Osmotic water permeability was also reported to be cAMP-independent and independent of osmotic gradient size and direction. As in bile duct cholangiocytes, a functional homolog of the gallbladder epithelium, subapical AQP1 was suggested to translocate to the apical membrane to secrete water. Based on its subcellular pattern of localization, gallbladder AQP8 was instead speculated to mediate the absorption of water and, to a lesser extent, to contribute to secreting water into the lumen [36] (Figure 4).

Nevertheless, the exact physiological relevance of AQP1 and AQP8 in gallbladder function remains a debated topic. Discrepant results are reported in the literature. A study found similar BA concentrations in gallbladders from wild-type and AQP1-ablated mice with no apparent functional substitution of AQP1 by AQP8 [103]. This observation, however, was not consistent with previous works reporting a temporal association between diminished gallbladder concentrating function and decreased AQP1 or AQP8 levels [101], and the results were obtained with leptin-deleted mice submitted to leptin replacement where the hormone altered the gallbladder volume acting on the AQP-mediated absorption/secretion of water [104]. Further and more targeted work is therefore required to clarify the question.

## 3. The Role of AQPs in Hepatobiliary Diseases

Several diseases affecting the hepatobiliary tree are associated with abnormal bile fluid transport and cholestasis [24,105]. Deranged hepatobiliary AQPs and bile secretion exist in experimental models of cholestasis, and studies with cellular and murine models of gallstone disease suggest an association between altered cholangiocytes’ AQP expression/localization and gallbladder concentrating function.

### 3.1. Liver and Bile Ducts

AQPs play a significant role in the development and progression of hepatocellular carcinoma (HCC). AQP3 is upregulated, while AQP7 and AQP9 are downregulated in HCC, with AQP3 expression correlating with aggressive tumor features [106]. A study showed that AQP3 is upregulated in HCC tissues and inversely correlated with miR-124 expression [107]. AQP9 shows altered expression and localization, dependent on liver pathology, with reduced levels in HCC tissues [108]. The loss of AQP8 and AQP9 contributes to apoptosis resistance in HCC [109]. AQP5 is linked to tumor invasiveness, although its prognostic impact remains unclear [110].

AQPs have an important role in other liver diseases and injuries. *Aqp9* knockout (KO) mouse models demonstrated that silencing Aqp9 reduced hepatic lipotoxicity, providing protection against subsequent inflammation, oxidative stress, apoptosis, and pyroptosis [111]. In leptin-deficient (ob/ob) mice, which are used as a model for NAFLD, fasting led to reduced AQP9 expression and function, along with higher plasma glycerol levels compared to lean mice. This suggests that AQP9 may play a role in liver steatosis [112]. Similarly, reduced AQP9 expression has been observed in liver biopsies from morbidly obese patients undergoing bariatric surgery, which has been proposed as a potential protective mechanism against further fat accumulation in the liver [25].

In a cell model of NAFLD induced by oleic acid in LO2 cells, overexpression of AQP9 worsened steatosis, while silencing AQP9 reduced it [113]. Consistent findings were reported in a HepG2 cell model. Additionally, treatment with oleic acid increased p38 phosphorylation, and blocking p38 prevented AQP9 upregulation, suggesting that AQP9 contributes to oleic acid-induced hepatic steatosis in HepG2 cells through p38 signaling [114].

Emerging data suggest the role of AQP1 in arterial capillary proliferation in the cirrhotic liver. AQP1 was primarily found in the proliferating arterial capillaries in human cirrhotic and late-stage primary biliary cirrhosis (PBC) livers, suggesting that AQP1 may trigger angiogenic responses. This could increase arterial blood flow into the sinusoids, raising sinusoidal microvascular resistance and contributing to the worsening of portal hypertension in cirrhosis [115].

Use of experimental models of cholestasis including extrahepatic obstructive cholestasis [116], estrogen-induced cholestasis [56], and sepsis-induced cholestasis [117] suggest that the dysregulated expression of AQP8 at the canalicular side of hepatocytes contributes to the development of cholestasis [18]. The role of AQP8 in cholestasis is indicated by the downregulation of canalicular AQP8 and decreased canalicular osmotic water permeability [56,116]. Deranged hepatocyte solute transporters and AQP8 function are likely to impair the coupling of osmotic gradients and canalicular water flow. Such findings suggest that cholestasis can also result from a mutual occurrence of reduced solute transport and impaired water permeability [118]. Notably, the adenoviral transfer of human AQP1 gene to rat liver improved bile flow in estrogen-induced cholestasis, with potential therapeutic implications for cholestatic diseases [119]. Matsumoto and colleagues found that the reduction of the paracellular or transcellular canalicular water flow has no significant effect on BA excretion [57]. These findings support novel hypotheses about the onset and progression of cholestasis [120,121]. AQP9, for example, is the basolateral AQP contributing to water movement from the sinusoidal blood into the hepatocyte. Of note, AQP9 was seen to be downregulated post-transcriptionally in a rodent model of extrahepatic cholestasis [64]. The reduced bile flow, biliary sludge, and cholestasis accompanying human hepatic ischemia and hypoxia were shown to involve a reduction in the hepatocyte AQP8 protein through a mechanism occurring at a post-translational level [122].

Hepatic cystogenesis associated with altered expression and subcellular localization of AQP1 (together with CFTR and AE2) was found in a rat model of autosomal recessive polycystic kidney disease. Liver cysts likely grow due to the increased fluid accumulation triggered by the overexpression and ectopic localization of AQP1, CFTR, and AE2 in cystic cholangiocytes [123]. Disruption of the *Aqp11* gene in mice led to intracellular vacuolization of periportal hepatocytes. This transgenic mouse developed a severe form of polycystic kidney disease (PKD) with uremic death before weaning due to renal failure [33]. Since the life span of *Aqp11*^−/−^ mice was limited by kidney disease where cysts were generated from intracellular vacuolization of proximal tubular cells, the liver phenotype could be premature. Polycystic livers are expected in *Aqp11* knockout mice since cysts are often seen in the biliary epithelia of PKD patients and mice [124]. Further research is required to assess whether the PKD caused by the depletion of AQP11 in mice leads to the same liver cysts induced by the autosomal recessive form of PKD, a well-characterized form of PKD caused by the homologous *Cpk* gene.

The role of AQP3 in the pathogenesis of liver injury is poorly understood. AQP3-mediated intracellular H_2_O_2_ transport is required for NF-κB activation. The monoclonal antibody for AQP3 inhibition is emerging as a potential therapeutic approach for liver injury [125]. However, AQP3 was found to be the only aquaglyceroporin present in hepatic stellate cells (HSCs). HSC activation (e.g., in fibrosis) is associated with decreased AQP3 expression [126]. In extrahepatic cholangiocarcinoma, AQP3 is associated with its pathogenesis and severity [127].

### 3.2. Gallbladder

Gallstones are solid conglomerates growing in the biliary tree such as the gallbladder, e.g., cholecystolithiasis [128]. Symptoms can appear within 5 years in about 10% of patients and within 20 years in about 20% of patients [128,129,130]. Shifting from simple cholecystolithiasis to gallstone disease will increase the risk of developing recurrent symptoms. In addition, gallstone disease is one of the most prevalent and costly digestive diseases in Western countries, with a 20% prevalence in adulthood [128,131]. About 80% of the gallstones are made of cholesterol [132], while the remainder are pigment stones that contain less than 30% cholesterol. The prevalence of gallstones increases with age, is higher in women than men, and is associated with multiple risk factors [128,133] including insulin resistance, type 2 diabetes, expansion of visceral adiposity due to overweight and obesity, and metabolic syndrome [134]. As recently debated, gallstones can be largely considered a metabolic dysfunction-associated gallstone disease [130]. The pathogenesis of cholesterol cholecystolithiasis requires the interaction of several contributing factors and at least five primary defects. These include the hepatic hypersecretion of biliary cholesterol, leading to bile supersaturated in cholesterol and prone to the precipitation of solid cholesterol; genetic factors; and Lith genes over a background predisposing the patient to cholesterol gallstones acting at various levels [135,136,137], rapid phase transitions of biliary cholesterol with the formation of early solid, filamentous, and arc- and needle-like anhydrous cholesterol and later of thermodynamically stable plate-like monohydrate crystals [138,139,140,141,142]. Gallbladder stasis plus mucin hypersecretion and mucin gel accumulation in the gallbladder lumen are additional promoting factors for cholelithogenesis as well as immune-mediated gallbladder inflammation. Gallbladder dynamics can be disrupted during cholesterol lithogenesis, as supersaturated hepatic bile delivers large amounts of solubilized cholesterol to the gallbladder epithelial cells. Here, cholesterol is converted to cholesteryl esters and stored in the mucosa and lamina propria, and excess cholesterol in the smooth muscle plasmalemma will stiffen the smooth muscle membrane and impair the CCK-1 receptor signaling cascade. Chronic gallbladder inflammation [143] and oxidative stress [144] are additional factors. Intestinal factors account for the increased transport of cholesterol from the intestinal lumen to the liver and intestine where biotransformation by the resident colonic microbiota of primary to secondary bile acids occurs, with increased levels of biliary DCA. This BA is highly hydrophobic, promoting further hepatic cholesterol hypersecretion and cholesterol crystallization.

Reduced expression of the gallbladder AQP1 and AQP8 associated with a decrement of the gallbladder concentrating ability was found during lithogenesis in C57L mice susceptible to diet-induced cholesterol gallstones [101]. In line with this observation, recent work with AQP8-depleted mice showed accelerated gallstone formation, which was rescued by the adenoviral-mediated liver expression of AQP8 or AQP1 [18]. The same work reported a small molecule, scutellarin, that increased the hepatocyte expression of AQP8 both in vitro and in vivo. Interestingly, in *Aqp8*^+/+^ wild-type mice, scutellarin significantly increased bile formation, reduced bile lipid concentrations, and prevented cholelithiasis compared to *Aqp8*^−/−^ knockout mouse littermates.

Impaired mRNA levels of AQP1 and AQP4 were found in the gallbladder of leptin-deficient obese [Lep(ob)] mice undergoing leptin replacement [102]. Besides showing the characteristic obesity, Lep(ob) mice showed enhanced gallbladder volumes and diminished gallbladder contractility, the latter reflecting gallbladder stasis [104]. An immunohistochemical study aimed at evaluating whether similar phenomena also exert a role in cholesterol gallstone disease in humans did not find a significant relationship between AQP1 and AQP8 expression in the gallbladder epithelium and the presence or absence of gallbladder stones [100]. Further work is needed to better address the question also in light of a recent study with thyroid hormone (TH)-deficient and cholestatic mice showing sex-specific expression and localization of hepatobiliary AQPs, with lower cholesterol gallstone prevalence in female C57BL/6J mice [145]. The lower expression of hepatobiliary AQPs was associated with the reduced biliary water transport in male C57BL/6J mice by possibly contributing to the sex-dependent cholesterol gallstone prevalence seen under TH deficiency.

In gallbladder cancer (GBC), the role of AQPs, particularly AQP5, is gaining attention due to their potential involvement in promoting cancer cell growth, invasion, and metastasis. However, despite their known significance in other cancers, there is limited evidence regarding the specific mechanisms and clinical relevance of AQPs in gallbladder cancer. Early studies suggest that AQP5 may influence tumor behavior and patient prognosis, but further research is needed to clarify the broader role of AQPs and their potential as therapeutic targets in GBC [146,147].

## 4. Targeting AQPs with Novel Modulators?

Progress in developing drugs targeting AQPs has been limited, perhaps due to several assumptions, such as the perception that the AQP pore is inherently resistant to drug targeting, amplified by challenges in reproducing current experimental methods [148]. Nevertheless, AQP-targeted drug development represents a rapidly growing area in recent years, also thanks to promising acquisitions made with both natural compounds and synthetic compounds capable of selectively blocking the channel or modulating the expression or regulatory mechanisms of AQPs [149,150,151,152]. Discovering novel drugs aimed at AQPs to address conditions marked by disrupted water and solute balance will fulfill an urgent clinical demand given the current absence of pharmacological remedies [153,154].

Preclinical studies including cellular and animal studies showed that different synthetic and natural compounds can modulate AQPs in different disease models [78,151,155,156]. Valuable preventive and therapeutic strategies could derive from AQP modulators potentially able to target the expression of AQP1 and AQP8 in the liver, in cholangiocytes, and in the gallbladder epithelium. The subsequent modulation of bile concentration should be particularly useful in clinical conditions characterized by cholestasis [18,118,119,157] (e.g., cholestatic liver diseases, biliary cirrhosis, cholangitis) but also in subjects at high risk of gallstone formation. As depicted in Figure 5, targeting AQP1 and AQP8 could potentially help to improve bile flow, possibly decreasing symptoms and local inflammation. In this context, targeting AQP1 and -8 in diseases affecting bile secretion and flow could be an effective strategy to improve water homeostasis and regulate bile volume.

Further beneficial effects could derive from the activation of the nuclear receptor FXR as an effect of AQP8 overexpression. According to results from animal studies, this pathway should affect lipogenesis, improving fat overstorage in the liver in subjects with steatosis [49].

Stimulating clues also derive from preliminary observations pointing to a beneficial effect of high AQP1 and AQP5 expression in subjects with biliary tract cancer, mainly in terms of longer survival, smaller tumor size, and depth of tumor invasion [147]. These results, however, should be considered with caution, since further data suggest an association between AQP overexpression and pro-carcinogenic effects in several types of cancer, such as skin cancer [158], breast cancer [159], gastric cancer [160], and colon cancer [161]. Conversely, knockdown of AQPs can suppress tumorigenesis in vivo and inhibit the proliferative, migratory, and self-renewal capability of cancer cells [160,161,162].

## 5. Conclusions and Perspectives

Although recent studies have proposed clinically relevant AQP-targeted therapies, such as the development of AQP inhibitors or modulators, clinical trials are still lacking and there are many difficulties. The majority of AQPs are expressed in the digestive system and have important implications for the physiopathology of the gastrointestinal tract. In the hepatobiliary system, AQPs play a crucial role in maintaining the delicate balance of bile composition and flow and bile water secretion and reabsorption, as well as in plasma glycerol uptake by the hepatocyte and its conversion to glucose during starvation. Accumulating evidence suggests that AQPs may also be involved in canalicular and ductal bile secretion, gluconeogenesis, and microbial infection, and they may have other novel roles that affect liver function. Evidence from basic and translational studies documented that dysregulation of AQP expression or function is associated with various hepatobiliary disorders, including cholestatic liver diseases and cancer.

Recent advancements in our understanding of AQP biology and physiology have led to the identification of potential therapeutic targets and possible side effects. Small molecule modulators are being explored to restore or regulate AQP activity, thereby mitigating aberrant bile secretion patterns observed in hepatobiliary-related diseases. Addressing the root causes of bile secretion-related disorders, the use of AQP1 and AQP8 modulators could potentially help to improve bile flow and could offer novel therapeutic strategies. However, further basic research and clinical trials are still necessary to translate these promising therapeutic options into effective treatments.

## Figures and Tables

**Figure 1 ijms-25-12133-f001:**
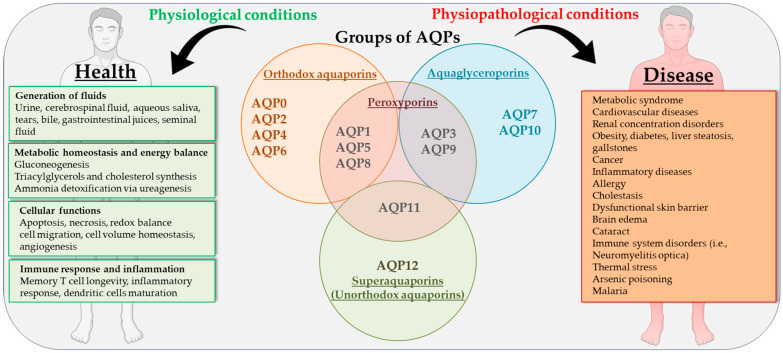
Humans possess thirteen aquaporins (AQPs) that have been identified as “orthodox aquaporins”, “aquaglyceroporins”, “peroxiporins”, and “superaquaporins”. AQPs have a pleiotropic role in health and disease.

**Figure 2 ijms-25-12133-f002:**
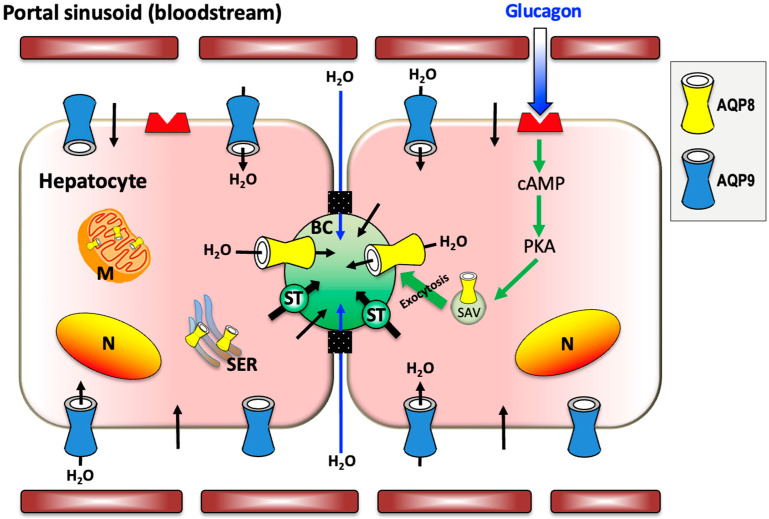
Proposed mechanism of AQP-mediated water diffusion in canalicular bile formation and secretion in hepatocytes. AQP8 facilitates the osmotic secretion of water into the bile canaliculus across the canalicular membrane, whereas AQP9 contributes to the diffusion of osmotic water from the sinusoidal blood into the cell. at levels considerably lower than those through the AQPs, water also moves through the phospholipid bilayer by simple diffusion (black arrows). The AQP-independent movement of water also occurs through the paracellular route across the tight junctions (blue arrows). Glucagon, a choleretic hormone, stimulates the microtubule-dependent canalicular targeting of AQP8-containing subapical vesicles to the canalicular membrane (green arrows). AQP8 is also present in mitochondria and smooth endoplasmic reticulum, where it is suggested to play other roles than that of facilitating the canalicular secretion of bile water. AQP9 also acts as the main pathway for the import of lipolytic glycerol from sinusoidal blood. BC, bile canaliculus; M, mitochondrion; N, nucleus; PKA, protein kinase A; SAV, subapical vesicles; SER, smooth endoplasmic reticulum; ST, salt transporters. The figure is partially adapted from Calamita and Delporte, *Cells*, 2023 [35].

**Figure 3 ijms-25-12133-f003:**
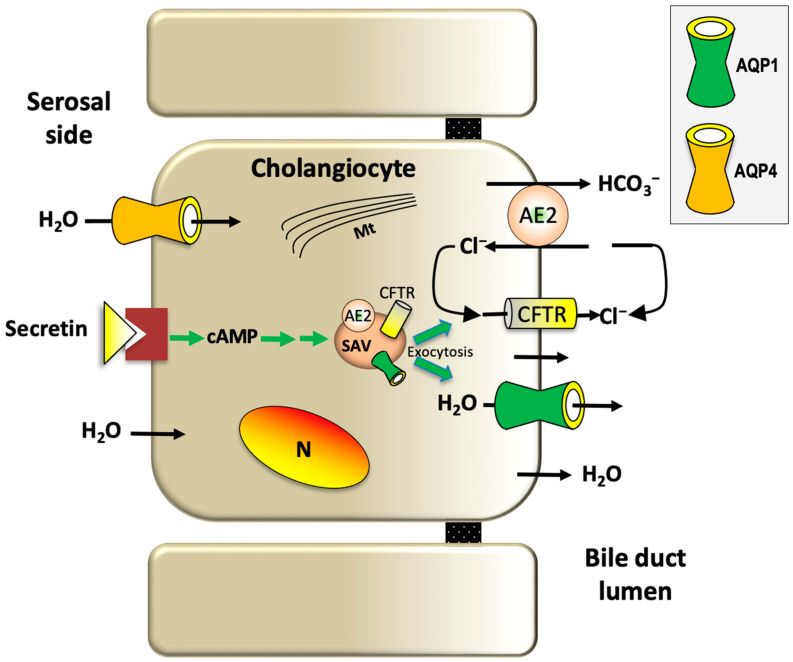
Proposed mechanism of AQP-mediated water movement in ductal bile secretion. Intrahepatic bile duct cholangiocytes. Secretin, through cAMP, induces the microtubule-dependent apical targeting and exocytic insertion of subapical vesicles containing AQP1 and CFTR Cl^−^ channels and the Cl^−^/HCO_3_^−^ exchanger AE2 into the apical plasma membrane. The exit of Cl^−^ via CFTR provides the luminal substrate driving the extrusion of HCO_3_^−^ into the lumen through AE2. HCO_3_^−^ and Cl^−^ ions generate the osmotic force driving the movement of water from blood plasma (mostly through basolateral AQP4) to the biliary lumen (through apical AQP1). AE2, anion exchanges isoform 2; CFTR, cystic fibrosis transmembrane conductance regulator; Mt, microtubules; N, nucleus; SAV, subapical vesicles. The figure is partially adapted from Calamita and Delporte, *Cells*, 2023 [35].

**Figure 4 ijms-25-12133-f004:**
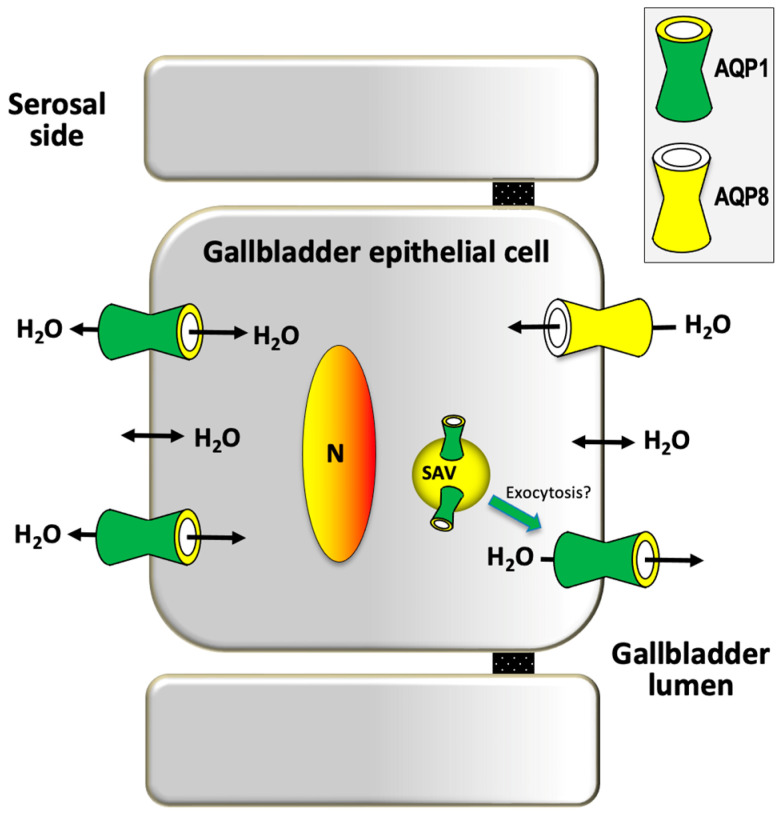
Proposed mechanism of AQP-mediated water movement in cystic bile absorption/secretion. Gallbladder epithelial cells. AQP8 and AQP1 mediate the osmotic absorption and secretion of water into/from the gallbladder lumen, respectively. AQP1 is also present in subapical vesicles (SAV) that are likely destined to be redistributed to the apical membrane by increasing the osmotic water permeability. Basolateral AQP1 facilitates the entry/extrusion of water into/out of the epithelial cells. The figure is partially adapted from Calamita and Delporte, *Cells*, 2023 [35].

**Figure 5 ijms-25-12133-f005:**
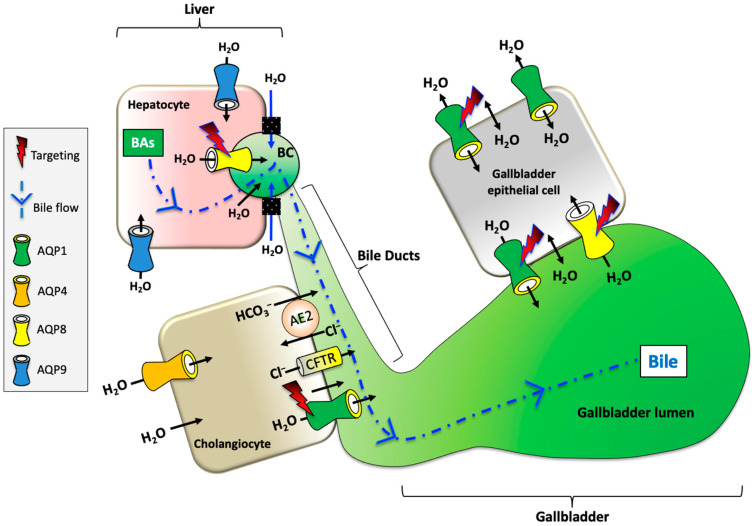
AQP-mediated water movement and bile flow in hepatocytes, cholangiocytes (bile ducts), and gallbladder epithelial cells. AQP1 and AQP8 are proposed as targets in diseases affecting bile secretion and flow (dotted blue arrow) to improve water homeostasis and regulate bile volume. BAs, bile acids.

**Table 1 ijms-25-12133-t001:** Localization and suggested physiological relevance of hepatobiliary aquaporins expressed at significant levels.

Hepatobiliary Section	Aquaporin	Cellular Location and Species	Subcellular Location	Functional Involvement
Liver parenchyma	AQP8	Hepatocytes (r, m, h)	APM, SAV, IMM, SER	Canalicular bile secretion; cytoplasmic osmotic homeostasis; mitochondrial ammonia detoxification and ureagenesis; mitochondrial H_2_O_2_ release; cholesterol biosynthesis; regulation of metabolic signaling
	AQP9	Hepatocytes (r, m, h)	BLM	Uptake of glycerol during starvation; lipid homeostasis; import of water from sinusoidal blood; catabolic urea extrusion
	AQP11	Hepatocytes (m)	RER	RER homeostasis; liver regeneration
Intrahepatic bile ducts	AQP1	Cholangiocytes (m, r, h)	APM, SAV, BLM	Secretion and reabsorption of ductal bile water
	AQP4	Cholangiocytes (m, r)	BLM	Secretion and reabsorption of ductal bile water
Gallbladder	AQP1	Epithelial cells (m, h)	APM, BLM, SAV	Cystic bile secretion/reabsorption
	AQP8	Epithelial cells (m, h)	APM, SAV	Cystic bile absorption (?)
Portal sinusoids; PVP; BV	AQP1	Endothelial cells (h)	APM, BLM	Bile formation and flow
Other hepatic cell types	AQP3	Kupffer cells (h)	PM	Cell migration and proinflammatory cytokines secretion (?)
	AQP8	Kupffer cells (r)	PM	Repopulation of Kupffer cells during liver regeneration (?)
	AQP3	Stellate cells (h)	PM	Adiponectin-mediated inhibition of hepatic stellate cells activation
	AQP11	Stellate cells (r)	Undefined	Control of activated hepatic stellate cells proliferation

Abbreviations: APM, apical plasma membrane; BLM, basolateral plasma membrane; BV, blood vessels; h, human; IMM, inner mitochondrial membrane; m, mouse; PM, plasma membrane; PVP, peribiliary vascular plexus; r, rat; RER, rough endoplasmic reticulum; SER, smooth endoplasmic reticulum; SAV, subapical membrane vesicles; ?, to be definitively confirmed.

## Data Availability

Not applicable.

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
