# Peer review of "Aquaporins in Biliary Function: Pathophysiological Implications and Therapeutic Targeting"

_ijms, 2024, doi:10.3390/ijms252212133_

Round 1

Reviewer 1 Report

Comments and Suggestions for Authors

This is a good review regarding Aquaporins (AQPs)s focusing its physiological and pathogenetic and role for liver, bile ducts, and gallbladder. These are the hot topic and thus this review would be useful for many researchers. The quality of English is fine, and the manuscript is well-written. I recommend publication of this review and could agree for publication of even present form. However, my minor comments and interests are following. Although the authors have described the role of AQPs for cholestatic diseases and cholelithiasis. I have found the research paper regarding AQP3 and liver injury (Nat Commun. 2020;11(1):5666). How about the role of AQPs for liver injury or hepatitis? I have also interest for the role of AQPs in malignancy such as hepatocellular carcinoma, cholangiocarcinoma, and gallbladder cancer. Please indicate “Author Contributons” in manuscript.

Author Response

Thank you very much for your thoughtful and encouraging feedback on our manuscript. We greatly appreciate your time and effort in reviewing our work. We would like to address your comments as follows:

1a. This is a good review regarding Aquaporins (AQPs)s focusing its physiological and pathogenetic and role for liver, bile ducts, and gallbladder. These are the hot topic and thus this review would be useful for many researchers. The quality of English is fine, and the manuscript is well-written. I recommend publication of this review and could agree for publication of even present form.

Answer. We are delighted to hear that you found our review valuable and that the topic resonates with current research interests. Your positive remarks on the manuscript’s quality and relevance are deeply appreciated.

1b. However, my minor comments and interests are following. Although the authors have described the role of AQPs for cholestatic diseases and cholelithiasis. I have found the research paper regarding AQP3 and liver injury (Nat Commun. 2020;11(1):5666). How about the role of AQPs for liver injury or hepatitis? I have also interest for the role of AQPs in malignancy such as hepatocellular carcinoma, cholangiocarcinoma, and gallbladder cancer.

Answer. We appreciate your suggestion to include further discussion on AQPs in liver injury, specifically regarding the study on AQP3 (Nat Commun. 2020;11(1):5666). Accordingly we added to the paper the following paragraph:

 Line 446-452: The role of AQP3 in the pathogenesis of liver injury is poorly understood. AQP3-mediated intracellular H2O2 transport is required for NF-κB activation. Monoclonal antibody for AQP3 inhibition is emerging as a potential therapeutic approach for liver injury[121]. However, AQP3 was found to be the only aquaglyceroporin present in hepatic stellate cells (HSC). HSC activation (e.g. in fibrosis) is associated with decreased AQP3 expression [122]. In extrahepatic cholangiocarcinoma, AQP3 is associated with its pathogenesis and severity[123]. We agree that the role of AQPs in liver injury and malignancies is a significant area of interest and will be happy to incorporate this aspect into our manuscript. Accordingly, we expanded our discussion to explore the role of AQPs in such as hepatocellular carcinoma, cholangiocarcinoma, and gallbladder cancer, as suggested.

Line 379-408: AQPs play a significant role in the development and progression of hepatocellular carcinoma (HCC). AQP3 is upregulated, while AQP7 and AQP9 are downregulated in HCC, with AQP3 expression correlating with aggressive tumor features [101]. A study showed that AQP3 is upregulated in HCC tissues and inversely correlated with miR-124 expression [102]. AQP9 shows altered expression and localization, dependent on liver pathology, with reduced levels in HCC tissues [103]. The loss of AQP8 and AQP9 contributes to apoptosis resistance in HCC [104]. AQP5 is linked to tumor invasiveness, although its prognostic impact remains unclear [105].

AQPs have an important role in other liver diseases and injuries. Aqp9 knockout (KO) mouse models demonstrated that silencing Aqp9 reduced hepatic lipotoxicity, providing protection against subsequent inflammation, oxidative stress, apoptosis, and pyroptosis [106]. In leptin-deficient (ob/ob) mice, which are used as a model for NAFLD, fasting led to reduced AQP9 expression and function, along with higher plasma glycerol levels compared to lean mice. This suggests that AQP9 may play a role in liver steatosis [107]. Similarly, reduced AQP9 expression has been observed in liver biopsies from morbidly obese patients undergoing bariatric surgery, which has been proposed as a potential protective mechanism against further fat accumulation in the liver [108].

In a cell model of NAFLD induced by oleic acid in LO2 cells, overexpression of AQP9 worsened steatosis, while silencing AQP9 reduced it [109]. Consistent findings were reported in a HepG2 cell model. Additionally, treatment with oleic acid increased p38 phosphorylation, and blocking p38 prevented AQP9 upregulation, suggesting that AQP9 contributes to oleic acid-induced hepatic steatosis in HepG2 cells through p38 signaling [110].

Emerging data suggest the role of AQP1 in arterial capillary proliferation in the cirrhotic liver. AQP1 was primarily found on the proliferating arterial capillaries in human cirrhotic and late-stage primary biliary cirrhosis (PBC) livers, suggesting that AQP1 may trigger angiogenic responses. This could increase arterial blood flow into the sinusoids, raising sinusoidal microvascular resistance and contributing to the worsening of portal hypertension in cirrhosis [111].

And line 515-522: In gallbladder cancer (GBC), the role of AQPs, particularly AQP5, is gaining attention due to their potential involvement in promoting cancer cell growth, invasion, and metastasis. However, despite their known significance in other cancers, there is limited evidence regarding the specific mechanisms and clinical relevance of AQPs in gallbladder cancer. Early studies suggest that AQP5 may influence tumor behavior and patient prognosis, but further research is needed to clarify the broader role of AQPs and their potential as therapeutic targets in GBC [143,144].

2. Please indicate “Author Contributons” in manuscript.

We apologize for this mistake,  we did now insert an “Author Contributions” section, as per your request.

Reviewer 2 Report

Comments and Suggestions for Authors

The review presented by the authors is an excellent summary of the definition, classification and role of aquaporins in the pathophysiology of liver tissue.

The English is fluent, the sentences simple and well-calibrated.

I have no major comments for the authors

I personally consider the manuscript ready for publication

Author Response

The review presented by the authors is an excellent summary of the definition, classification and role of aquaporins in the pathophysiology of liver tissue.

The English is fluent, the sentences simple and well-calibrated.

I have no major comments for the authors

I personally consider the manuscript ready for publication

Answer. Dear Reviewer, thank you very much for your kind and supportive feedback on our manuscript. We are thrilled that you found the review to be a valuable and well-written summary of the role of aquaporins in liver pathophysiology. Your positive remarks on the clarity and fluency of the language are deeply appreciated.

We are grateful for your endorsement for publication, and we are pleased to know that you consider the manuscript ready in its current form.

Reviewer 3 Report

Comments and Suggestions for Authors

There are some comments.

It would be better to replace Fig. 1 with a clearer photo.

Please include “Figure 4” in the text.

It would be better to explain the role of SAV in Figure 4.

It would be better to further explain the impact of altered AQP expression in many diseases listed in Figure 1 and how these changes affect disease progression and prognosis.

Comments on the Quality of English Language

Please check English grammar and spelling.

Author Response

There are some comments.

Thank you very much for your helpful and detailed feedback on our manuscript. We greatly appreciate your thoughtful suggestions and will address each point as follows:

  1. It would be better to replace Fig. 1 with a clearer photo.

Answer. We thank the reviewer for this observation. We revised and replaced Figure 1 with a clearer image to enhance the visual quality and ensure the information is more accessible to readers.

  1. Please include “Figure 4” in the text.

Answer. We acknowledge the omission of “Figure 4” in the main text. The figure in question is now properly referenced and integrated.

  1. It would be better to explain the role of SAV in Figure 4.

We appreciate your suggestion to elaborate on the role of SAV in Figure 4. We included additional explanations both in the main text of the manuscript and in the figure legend.

Line 353-355:

Main text. ..and over subapical vesicles that can be incorporated into the apical membrane by means of a microtubule-dependent, cAMP-stimulated mechanism whose stimulation remains is not yet known [96].....

Figure legend. AQP1 is also present in subapical vesicles (SAV) that are likely destined to be redistributed to the apical membrane by increasing the osmotic water permeability.

  1. It would be better to further explain the impact of altered AQP expression in many diseases listed in Figure 1 and how these changes affect disease progression and prognosis.

Thank you for your valuable feedback. While the primary aim of our paper is to focus on the role of AQPs in the hepatobiliary tract and not to provide an in-depth discussion of their involvement in other diseases, we acknowledge the relevance of your suggestion. To address this, we have inserted additional references and integrated a brief paragraph that highlights the impact of altered AQP expression in various diseases, as listed in Figure 1. However, we have kept the focus aligned with our main objective of exploring AQPs in the hepatobiliary system.

Line 67-74: As shown in Figure 1, AQPs play a crucial role in maintaining metabolic homeostasis, with their distribution, expression, and function being regulated under both normal and pathological conditions. AQPs are involved in various physiological processes across multiple organ systems, and their dysregulation has been implicated in a range of diseases[14]. These include metabolic syndrome [15], cardiovascular diseases [16], renal concentration disorders [17], obesity [18], diabetes [19], liver steatosis [20], and gallstones [21]. Furthermore, AQPs are linked to cancer, inflammation, and a wide range of diseases [22-26].

  1. Please check English grammar and spelling.

We carefully reviewed the manuscript for any grammatical or spelling errors and make necessary corrections to improve the overall clarity and readability. All reviewed parts are in red.

Round 2

Reviewer 3 Report

Comments and Suggestions for Authors

The manuscript was well-revised.

1. It would be better to change as follows.

   i.e.  Neuromyelitis optica" ->e.g., Neuromyelitis optica

2. Please write the abbreviations r, m, and h in Table 1.

3. Fig. 2, 3, and 4 seem to be similar to pictures in other literature. 

   If the figures were modified, you can include a note like:

   "Modified from [Author(s), Year, Journal or Publisher". 

Comments on the Quality of English Language

Please check English grammar and spelling.

Author Response

Comment 1: It would be better to change as follows.   i.e.  Neuromyelitis optica" ->e.g., Neuromyelitis optica

Response: Thank you for this suggestion. We have updated the phrasing from "i.e., " to "e.g., as advised to better clarify the intent of the sentence.

Comment 2: Please write the abbreviations r, m, and h in Table 1.

Response: Thank you for noting this. We have added definitions for the abbreviations "r," "m," and "h" in Table 1 to ensure clarity for readers.

Comment 3: Fig. 2, 3, and 4 seem to be similar to pictures in other literature.  If the figures were modified, you can include a note like: "Modified from [Author(s), Year, Journal or Publisher". 

Response: Thank you for your observation regarding Figures 2, 3, and 4. These figures were partially adopted from our recent publications, but they have been redrawn and modified to fit the specific concepts and objectives of the current paper.

To acknowledge this adaptation, we have added a note below each figure “The figure is partially adapted from Calamita&Delporte, Cells, 2023 [36].”